# A New Dawn for the Use of Artificial Intelligence in Gastroenterology, Hepatology and Pancreatology

**DOI:** 10.3390/diagnostics11091719

**Published:** 2021-09-19

**Authors:** Akihiko Oka, Norihisa Ishimura, Shunji Ishihara

**Affiliations:** Department of Internal Medicine II, Faculty of Medicine, Shimane University, Izumo 693-8501, Shimane, Japan; ishimura@med.shimane-u.ac.jp (N.I.); si360405@med.shimane-u.ac.jp (S.I.)

**Keywords:** machine learning, deep learning, endoscopy, ultrasonography, radiology

## Abstract

Artificial intelligence (AI) is rapidly becoming an essential tool in the medical field as well as in daily life. Recent developments in deep learning, a subfield of AI, have brought remarkable advances in image recognition, which facilitates improvement in the early detection of cancer by endoscopy, ultrasonography, and computed tomography. In addition, AI-assisted big data analysis represents a great step forward for precision medicine. This review provides an overview of AI technology, particularly for gastroenterology, hepatology, and pancreatology, to help clinicians utilize AI in the near future.

## 1. Introduction

Rapid developments in artificial intelligence (AI) technologies bring huge benefits to daily life through smartphones (iPhone’s Siri, etc.), wearables (smart watches, etc.), and robotic assistants (smart speakers, self-driving cars, etc.) [1,2]. In the medical field, AI also holds great promise. Major advances in medical AI have had a tremendous impact at two main levels: (1) image recognition and (2) big data analysis. AI can detect very small changes that are difficult for humans to perceive. For example, AI can detect lung cancer up to a year before a physician [3], and AI can correctly diagnose skin cancer with superior diagnostic performance compared to that of a physician [4]. In addition, AI can reach the desired output within seconds and with more “consistent” performance. Doctors may have “inconsistent” performance due to insufficient training or exhaustion from busy clinical demands. A visual assessment by imaging physicians is qualitative, subjective, and prone to errors, and subject to intra-observer and inter-observer variability. AI may have better performance than physicians in some cases [5], and it has great promise to reduce clinician workload and the cost of medical care. However, it is necessary for clinicians to verify the output from AI for patient care.

In addition to image analysis, big data analysis is suitable for AI to generalize across a variety of data types and to provide interpretation across complex variables [6]. Therefore, AI techniques have been widely applied to big data analyses, such as in genomics, novel medicine discoveries, and predictions of disease outcomes [7,8,9]. For example, IBM Watson supports oncologists by providing possible therapeutic options based on information from over 300 medical journals, over 200 academic books, and over 15,000,000 pages of literature related to 11 types of neoplasia [10,11]. In the field of gastroenterology, AI has also made remarkable progress, and many international meetings highlight AI-related sessions. In addition, several new conferences have been established over the past few decades, such as the Global GI-AI Summit [12]. Owing to the potential for image recognition and big data analysis, not only clinician, but also researchers can benefit from the application of AI methodologies. This review focuses on recent AI research in the fields of gastroenterology, hepatology, and pancreatology (summarized in Figure 1) and provides an overview of AI technology to help clinicians utilize AI in the near future.

## 2. Artificial Intelligence

AI is “a broad discipline with the goal of creating intelligent machines, as opposed to the natural intelligence that is demonstrated by humans and animals” (from the state of AI report 2020) [1]. In 1950, Alan Turing published a landmark paper describing the creation of machines that “think” [13]. In 1955, John McCarthy et al. used the word “artificial intelligence” for the first time in a proposal for the Dartmouth Conference held in 1956 [14], which is considered the dawn of AI technology. In 1959, Arthur Samuel developed an algorithm for machine learning, a subfield of AI, which referred to a computer’s ability to learn from data in order to detect patterns and make decisions without explicitly being programmed for the output [15,16]. Before learning algorithms were developed, humans alone were required to analyze data and program machines with human-designed algorithms. In contrast, AI can automatically detect patterns and attributes from data and make decisions without human input. 

An integral breakthrough in AI technology came in 2012, when deep learning, a new type of machine learning, was developed by Geoffrey Hinton et al. [17]. The authors presented a dramatically improved error rate for visual recognition at a competition conference, the ImageNet Large Scale Visual Recognition Challenge (ILSVRC), jointly held by multiple universities in the United States [18]. Hinton’s team at the University of Toronto used deep learning for the first time to improve the error rate by about 10%. The network used was a convolutional neural network (CNN) called AlexNet, which has since been widely applied for image recognition tasks [19]. Deep learning uses a system called a neural network, which imitates the neuronal network of the human brain and combines different mathematical models. The input layer and the output layer are not sufficient to process complex information (Figure 2A), and more sophisticated analyses can be performed by creating intermediate layers between them. This increase in the number of intermediate layers is expressed as deep = deep, and deep learning is a computer processing system that has many such intermediate layers (Figure 2B). The layer is composed of a filter that extracts features from the original images to determine the characteristics of the original images where higher level features are extracted from lower level ones: for example, the first layer extracts patterns at the texture level, the second layer extracts patterns at the frame level, the third layer extracts at the shape level, and the last layer indicates a list of parts in the original input image. Notably, the filter is automatically created after recognition of the features through learning from the input data (see details and examples in [18,20,21,22,23]. This breakthrough in deep learning was facilitated by advances in graphic processing units (GPUs), which were faster than central processing units (CPUs) for real-time graphics and multitasking [18]. In 2015, AI outperformed humans in the ILSVRC. Another example to illustrate the outstanding performance of deep learning was indicated by AlphaGo, a deep learning algorithm to win the game “Go” [24]. These attractive developments in deep learning have greatly contributed to the proliferation of studies, which have attempted to automate the interpretation and evaluation of medical images and clinical data, and have expanded the application of AI to various fields. Indeed, over 10,000 papers in the medical field were published last year (Figure 2C). Based on these recent developments in AI technology, the U.S. Food and Drug Administration (FDA) enacted a law to approve medical AI devices in December 2016. In April 2018, the first AI device was approved to provide screening decisions without the assistance of a clinician’s interpretation for diabetic retinopathy in adults with diabetes [25]. To date, several AI-aided devices have been approved by the FDA and the European Union (EU) in the field of gastroenterology, hepatology, and pancreatology (Table 1).

## 3. Pharyngeal Cancer

Pharyngeal cancer is generally detected by otolaryngologists, and the majority of pharyngeal cancer patients are diagnosed at an advanced stage, resulting in a poor disease prognosis [26]. Therefore, early detection is critical to improve the survival rate of pharyngeal cancer patients. With recent advances in endoscopic technology, such as narrow-band imaging (NBI) and magnifying endoscopy, not only otolaryngologist, but even gastrointestinal endoscopist, are able to detect laryngopharyngeal cancers at an early stage [27,28,29]. A few reports regarding the AI-aided detection of pharyngeal cancers have been published by nasopharyngiologists and gastroenterologists [30,31,32,33]. Tamashiro et al. trained an AI-aided endoscopy system with 5403 images of superficial and advanced pharyngeal cancers and validated the system with 1912 images of cancers and non-cancers [32] (Figure 3A). The AI system correctly detected all cancers (even those smaller than 10 mm), and the pictures from NBI provided to the AI resulted in a much higher sensitivity (85.6%) than that from white-light endoscopy (70.1%). In actual clinical care situations, “real-time” detection is more practical and effective. Kono et al. developed a real-time detection system [33,34] that diagnosed 23/25 pharyngeal cancers as cancers (sensitivity: 92%) and 17/36 non-cancers as non-cancers (specificity: 47%) in a validation study, which used video images with a high transaction speed of 0.03 s per image. They theorized that the pseudo-positive or negative cases were due to the complex environment of the laryngopharyngeal area, including things such as saliva, bubbles, blurring, and inadequate filming conditions. Further improvements in the AI system with a variety of training images from normal and pharyngeal cancer patients are needed.

## 4. Upper Gastrointestinal Diseases

The overall survival rates for upper gastrointestinal cancers are poor, since many are diagnosed at advanced stages [38]. However, if detected early, the five-year survival rates exceed 90% [39,40,41]. For the early detection of neoplasms, endoscopists should pay attention to very small changes in the mucosa. Unfortunately the detection rates and accuracy of endoscopic diagnosis depends largely on the endoscopists’ experience [42]. AI-aided detection systems are therefore a hopeful and promising tool in this field.

### 4.1. Esophageal Cancer

Esophageal cancer is the seventh most common neoplasm and sixth most deadly cancer worldwide [43]. Squamous cell carcinoma (SCC) is the most common tumor type of all esophageal cancers [43], and several AI systems to detect SCCs have been reported [44,45,46,47,48]. Recently, Tokai et al. demonstrated that AI using a CNN detected 95.5% (279/291) of SCCs in 10 s [44]. They also showed that NBI was more sensitive than white-light imaging, which is consistent with previous reports [45]. In addition, they demonstrated that the AI correctly estimated the invasion depth with a sensitivity of 84.1% and an accuracy of 80.9%, which was higher than that of endoscopists. In addition, several reports showed that magnified endoscopy enhanced the accuracy of the depth diagnosis [46,47]. Interestingly, the AI performance was the same as that of the experts. In clinical endoscopy practice, ‘real-time’ diagnosis is required for AI-aided endoscopy. In a multicenter case-control study, Luo et al. validated an established AI system, known as GRAIDS, which was trained with 1,036,496 endoscopic images, and demonstrated high sensitivity, specificity, and accuracy [48]. This is one of the largest studies in the field of AI for medical applications. 

While the majority of esophageal cancers are SCCs, the incidence of adenocarcinoma in the esophagus is increasing rapidly in Europe and North America [49]. Several reports have been published for the detection of adenocarcinoma using AI methodologies [50,51,52,53,54,55,56,57,58]. An AI system with white-light endoscopy developed by de Groof et al., detected Barrett’s neoplasia with high performance (a sensitivity of 95%, a specificity of 85%, and an accuracy of 92%) [50]. Subsequently, they developed an AI algorithm with multi-step training and successfully improved the accuracy of AI detection for Barrett’s neoplasia over the performance of endoscopists [51]. Recently, Hashimoto et al. used a high-speed real-time AI detection algorithm and demonstrated high sensitivity (96.4%), specificity (94.2%), and accuracy (95.4%) for the detection of early neoplasia on Barrett’s esophagus [52]. A recent meta-analysis by Arribas et al. showed that AI-aided endoscopy can detect both types of esophageal neoplasia, SCC and adenocarcinoma, with high sensitivity (approximately 90%) and accuracy (AUC approximately 0.95) [53], indicating that an AI system is a promising tool to avoid missing neoplasia during endoscopy. 

### 4.2. Gastric Cancer

Gastric cancer is the fourth most lethal cancer worldwide [43]. As with the other gastrointestinal cancers described above, early detection is critical to improve survival rates [59]. In 2018, Hirasawa et al. first reported a novel AI-aided (computer-aided) diagnostic system for the detection of gastric cancer using a deep learning CNN [35] (Figure 3B). In total, 13,584 endoscopic images of gastric cancer as well as non-cancer images were collected to train the AI system. For verification of the diagnostic accuracy, 2296 endoscopic images of 69 consecutive cases of gastric cancer (77 lesions) were used. The trained AI detected 92.2% of gastric cancer lesions. Using another CNN algorithm, Wu et al. demonstrated higher performances in the AI group than those of expert endoscopists (accuracy 92.5% vs. 89.7%, sensitivity 94% vs. 93.9%, specificity 91% vs. 87.3%) [60]. In addition to these “still” image detection methodologies, Horiuchi et al. developed an AI to enable “real-time” diagnosis using magnifying endoscopy with NBI [61]. The AI system demonstrated an accuracy of 85.1%, a sensitivity of 87.4%, and a specificity of 82.8%, which was significantly more accurate than two experts. More recently, they employed a larger number of experts (67 endoscopists) to determine whether the performance of the AI detection system is better than that of endoscopists [62]. The AI system detected a greater number of early gastric cancer cases in a shorter time than the endoscopists with a significantly higher sensitivity of 58.4% versus 31.9%, respectively. Although the accuracy of the system was slightly lower than that of the experts, and requires further training and adjustments, it presents a promising tool to detect early cancer lesions. 

Since AI is highly sensitive in image recognition, there can be misdiagnoses. To improve the accuracy of the diagnosis for cancer versus non-cancer, several reports have been published. In a study by Hirasawa et al., most of the misdiagnoses by AI were gastritis diagnosed as gastric cancer mainly due to the high sensitivity of the AI [35]. Horiuchi et al. established AI-aided magnifying endoscopy using NBI and demonstrated that gastritis could be distinguished from gastric cancer with a correct diagnostic rate of 85.3% [63,64]. Another color-enhanced imaging modality, flexible spectral imaging color enhancement (FICE) can also be used for the AI-aided detection of gastric cancer. Miyaki et al. used a support vector machine, which includes machine learning with training and validation images, and found that the system yielded a detection accuracy of 85.9%, a sensitivity of 84.8%, and a specificity of 87.0% [65]. Furthermore, for gastritis, the delineation of cancerous regions can be challenging. Kanesaka et al. first reported the introduction of AI for the diagnosis of gastric cancer [66]. The diagnosis by AI showed relatively good results with a sensitivity of 65.5%, a specificity of 80.8%, and a correct diagnostic rate of 73.8%. Kubota et al. developed an AI for the diagnosis of the invasion depth using a neural network and demonstrated accuracies of 77%, 49%, 51%, and 55% for T1, T2, T3, and T4 stages, respectively [67]. Zhu et al. also developed an AI for the diagnosis of the invasion depth in gastric cancer. For the diagnosis, which can distinguish a depth of M, SM1, SM2, or deeper, for all gastric cancers, including advanced stages, the sensitivity, specificity, and accuracy for AI were 76.5%, 95.6%, and 89.1%, respectively, and for endoscopists, they were 87.8%, 63.3%, and 71.5%, respectively [68]. Yoon et al. reported an AI that could classify early gastric cancer into intramucosal or submucosal cancers, with an area under the curve (AUC) of 0.851 [69]. Furthermore, they found that the factor that contributed most to the AI prediction of tumor depth was histologic differentiation. Undifferentiated-type histology corresponded to a lower AI accuracy.

### 4.3. Helicobacter pylori Infection and Gastric Atrophy

Helicobacter pylori infection followed by gastric atrophy is an important cause of gastric cancer [70]. Early diagnosis and management of *H. pylori* infection and gastric atrophy is a key strategy to reduce gastric cancer-related death. However, the diagnosis of *H. pylori* infection based on endoscopic findings remains a subjective process, which greatly depends on the competence of the treating physician, and the accuracy of diagnosis varies widely [71]. Shichijo et al. first developed an AI system for the diagnosis of *H. pylori*-induced gastritis, using 32,208 white-light endoscopic images from 1768 patients both *H. pylori* positive and negative for training [72]. Interestingly, the AI exceeded the performance of the endoscopists to diagnose *H. pylori* infection. In addition, given that the detection of *H. pylori* infection includes current infection and successful eradication therapy (post-eradication), the authors [73] and another group [74] trained an AI system with cases that included current infection, no infection, and post-eradication. These studies demonstrated a similar diagnostic performance compared to that of endoscopists, with a correct diagnostic rate of 84.2% for no infection, 82.5% for current infection, and 79.2% for post-treatment resolution [74]. In a more recent study, Nakahira et al. developed a unique AI system to evaluate the risk of gastric cancer [75]. The AI was trained on images of high-risk (patients with gastric cancer), moderate-risk (patients with current or past *H. pylori* infection or gastric atrophy), or low-risk (patients with no history of *H. pylori* infection or gastric atrophy) patients. The trained system successfully stratified the risk of cancer for the low-, moderate-, and high-risk patients, who were diagnosed by the AI as having gastric cancer at 2.2%, 8.8%, and 16.4%, respectively. 

### 4.4. Upper Gastrointestinal Bleeding

In addition to image analysis above, AI can be applied to big data analysis to predict disease outcomes. For acute upper gastrointestinal bleeding, a systematic review by Shung et al., which included 14 studies with 30 assessments of machine learning models, revealed that AI performance was better than validated clinical risk scores to predict mortality from upper gastrointestinal bleeding [76]. Then, the authors published an excellent risk scoring system using machine learning models with a greater AUC, higher levels of specificity, and a 100% sensitivity compared to the clinical risk scores [77].

### 4.5. Quality Control

Blind spots potentially exist, even if endoscopists intend to observe the entire stomach, which is a cause of missed gastric cancer [78]. Wu et al. established a real-time quality improvement system, named WISENSE (wise + sense), and conducted a randomized controlled trial of 324 patients to confirm the comprehensiveness of the real-time imaging for the entire stomach. The study findings indicated that the AI reduced imaging omissions by 15% [60,79]. Using a similar AI, Chen et al. conducted a randomized controlled trial comparing six groups, including the presence or absence of sedation, normal-diameter or small-diameter endoscope, and with or without AI. They reported that normal-diameter endoscopy, with AI, and under sedation resulted in significantly fewer omissions [80]. An AI system should be able to detect cancer even under less than ideal conditions because suboptimal conditions are quite common in daily medical practice, particularly in pharyngeal areas. Normal images under such “real life” conditions are needed for AI training.

## 5. Gastrointestinal Stromal Tumor (GIST)

Large GISTs often show various findings on endoscopy and endoscopic ultrasonography (EUS), which makes it challenging for clinicians to distinguish GISTs from other submucosal tumors (SMTs). Minoda et al. reported the first study to evaluate the ability of AI to diagnose SMTs by EUS images. The AI-aided EUS showed a good diagnostic capability for large SMTs (≥20 mm) with a sensitivity of 91.7%, a specificity of 83.3%, and an accuracy of 90.0%, which were better than those of the EUS experts, (50.0%, 83.3%, and 53.3%, respectively) [81]. The AUC of the AI-aided EUS for large SMTs was 0.965, which was significantly higher than that of the EUS expert readers (0.684). In the future, with the help of the AI-aided EUS, non-experts might be able to make a differential diagnosis of GIST with the same or higher accuracy than that of EUS experts and without an invasive sampling process.

## 6. Duodenal and Small Intestinal Lesions

Duodenal neoplasia is relatively rare and sometimes missed during upper gastrointestinal endoscopy. Inoue et al. pretrained an AI system (deep learning CNN) with many cases of duodenal neoplasia (65 adenomas, 31 high-grade dysplasias) and showed that the system could detect duodenal neoplasia (sensitivity 94.7%), although there were some false positives (12.6%) probably due to a peristalsis-related raised fold [82]. The diagnostic ability of video capsule endoscopy (VCE) for small intestinal lesions is as high as 63%, which is superior to push endoscopy (single or double balloon endoscopy) [83]. VCE produces large amounts of data (over 50,000 images), which require considerable time for manual review by clinicians (30–120 min) [84,85]. Time-saving approaches are needed [86]. AI is a promising tool for this, and several studies have been performed and summarized previously [87]. Small intestinal bleeds are the most frequent indication for the use of VCE. Although commercially available reading systems include blood content enhancement algorithms, referred to as “suspected blood indicators” (SBIs), the false positive rate is still high at over 70% [88]. Xiao et al. and Hassan et al. developed AI algorithms for the detection of bleeds with high sensitivity and specificity (99%) [23,89]. Aoki et al. also developed a novel AI-based blood detection algorithm with high sensitivity, specificity, and accuracy (96.6%, 99.9%, and 99.8%, respectively), which were significantly higher than those of the SBI (76.9%, 99.8%, and 99.3%, respectively) [90]. They also showed the utility of an AI-based system for various small intestinal lesions (erosion, ulcer, angioectasia, and protruding lesions) in their multiple clinical studies [36,91,92] (Figure 3C). Hopefully, these novel AI algorithms will reduce the reading time for clinicians in the near future [93]. However, there are some limitations for developing AI-aided VCE, since small intestinal diseases are rather rare, and it is difficult to obtain sufficient large data sets for training. In addition, the VCE images may contain many artifacts (dark and red) and other objects (bile, food, air bubbles, etc.). There is a need for large collaborative databases to develop more precise systems. 

## 7. Colon Cancer and Polyps

Colorectal cancer is the second most lethal cancer worldwide [43]. The total removal of colorectal adenomas by colonoscopy (clean colon) can reduce colorectal cancer deaths by 53% [94]. It is well known that approximately 20–50% of colorectal polyps are overlooked [95,96]. This incidence might be affected by the skill and fatigue of the endoscopist. Recent developments in deep learning algorithms have improved the detection sensitivity and specificity of AI-aided colonoscopy (in other words, computer-aided detection (CADe)). Using a deep learning algorithm, Misawa et al. first reported real-time detection for colon polyps, with a sensitivity of 90% and a specificity of 63.3% [97]. Urban et al. improved the specificity to 93% with a sensitivity of 93% using a wider variety of images (4088 unique polyps) for training [98]. In a more resent study, they demonstrated that AI-aided colonoscopy trained by more images (56,668 images) detected polyps with a higher sensitivity (98%) and an improved specificity (93%) using a novel publicly accessible video database (entitled SUN-database: http://amed8k.sundatabase.org/ (accessed on 19 September 2021)) they established [99]. The first randomized, controlled trial was conducted by Wang et al., in which a total of 1058 patients (536 standard colonoscopies and 522 computer-aided colonoscopies) were included [100]. The AI-aided colonoscopy significantly increased the adenoma detection rate (53% in the AI group versus 31% in the control group). Recently, the same group conducted high-quality studies, including a double-blind randomized trial with an AI–colonoscopy system compared to a sham system, and demonstrated that the adenoma detection rate was significantly higher in the AI-colonoscopy group (34%, 165/484) than in the sham group (28%, 132/478) [101]. The adenoma miss rate was significantly lower in the AI–colonoscopy group compared to a routine colonoscopy (13.8% vs. 40.0%) [102]. They mentioned that the characteristic profiles of the polyps initially missed by the endoscopist but identified by the AI system were of small size, isochromatic, flat, and located behind the colon folds, as well as on the edge of the visual field. 

If optical colonoscopy is not possible, a colon capsule endoscopy or CT colonography may be performed. In a clinical trial, Deding et al. found that the sensitivity of colon capsule endoscopy (estimated location by AI) following an incomplete optical colonoscopy was superior to CT colonoscopy, and the relative sensitivity of colon capsule endoscopy compared with CT colonography was 2.67 for polyps >5 mm and 1.91 for polyps >9 mm [103].

To reduce unnecessary endoscopic resections and decrease complications and medical costs, it is important to distinguish neoplasms from non-neoplasms. In the first prospective clinical study in the field, Kominami et al. achieved high performance for a real-time diagnosis by an AI-aided colonoscopy (in other words, computer-aided diagnosis (CADx)), with a sensitivity of 93.3% and a specificity of 93.3% [104]. Tamami et al. demonstrated that a computer-aided NBI colonoscopy correctly diagnosed T1b stage cancer with a sensitivity of 83.9% and a specificity of 82.6%, which was better than a normal endoscopy [105]. Mori et al. successfully proved the utility of AI-aided endocytoscopy, which is an ultra-high magnification endoscopy that permits an in vivo assessment of cellular structure, in prospective clinical trials. In their studies, the AI-aided endocytoscopy had a sensitivity of 92% and an accuracy of 89.2%, which was quite similar to expert pathologists [106,107]. In a recent multicenter study, Kudo et al. showed a much better performance (96.9% sensitivity, 100% specificity, and 98% accuracy) of AI endocytoscopy trained using 69,142 endocytoscopic images, taken at 520× magnification, from patients with colorectal polyps who underwent endoscopy at five academic centers [108]. These tremendous efforts by endoscopists and engineers have resulted in a powerful basis for the development of AI-assisted devices, and several AI-aided endoscopic systems have been approved by the FDA and the EU (Table 1). Using AI-aided devices, endoscopists can begin an endoscopic exam immediately by connecting the endoscope to a terminal and monitor equipped with the software. Moreover, a prototype of a novel AI including a colonoscope, which has two lenses, a 160° to 240° angle lateral-backward-view lens and a standard 160°-angle forward-view lens, was published with videos included [109].

Depth prediction for colon cancer is another issue in a colonoscopy diagnosis. Takeda et al. demonstrated that AI endocytoscopy correctly diagnosed invasive colorectal cancer with a sensitivity of 98.1% and a specificity of 100% [110]. Chen et al. used EUS with AI for predicting tumor deposits with a higher AUC than that obtained by magnetic resonance imaging (MRI) [111]. Recently, Kudo et al. established an AI prediction system using patient’s data (age, sex, tumor size, morphology, lymphatic and vascular invasion, and histology), demonstrating that the AI system identified patients with lymph node metastases of T1 colon cancer better than the United States guidelines (AUC 0.83 vs. 0.73) [112]. They mentioned that these prediction models might be used to determine which patients require additional surgery after endoscopic resection of T1 colon cancer.

## 8. Inflammatory Bowel Disease

The incidence of inflammatory bowel disease (IBD), represented by Crohn’s disease (CD) and ulcerative colitis (UC), is increasing throughout the world, but its pathogenesis remains unclear [113,114,115,116]. Recent studies have indicated that IBD is a multifactorial immune-mediated disease resulting from a complex interplay between host genetic, environmental, and resident microbial factors [115,117,118,119]. To explore the pathogenesis, big data analysis by AI, such as pathological elucidation and biomarker identification, is ongoing and summarized in another review [120]. Using AI data analysis, Waljee et al. predicted remission in patients with moderate-to-severe CD with an AUC of 0.78 at week 8 and an AUC of 0.76 at week 6 [121]. Wang et al. applied AI to predict medication non-adherence in CD patients [122].

An endoscopic assessment of inflammation in IBD may vary among endoscopists depending on their level of experience. Several AI-aided UC scoring algorithms trained by unbiased UC imaging data that were linked to histological data demonstrated excellent performances in distinguishing endoscopic remission (Mayo 0–1) from moderate-to-severe disease (Mayo 2–3) [123,124,125]. Even Mayo 1 level mucosa has very mild inflammation. Ozawa et al. focused on distinguishing Mayo 0 from 0–1 and showed a high level performance of AI-aided diagnosis with an AUC of 0.86 and 0.98 for Mayo 0 and 0–1, respectively [126]. In a more resent prospective study, Takenaka et al. trained an AI algorithm with 40,758 images of colonoscopies and 6885 biopsy results from 2012 UC patients and showed that the system identified endoscopic remission with 90.1% accuracy and histologic remission with 92.9% accuracy [127]. Another approach using endocytoscopy with AI was reported by Maeda et al. [128]. As indicated above, using capsule endoscopy, Kumar et al. reported the first AI-aided diagnostic system for CD lesions with various levels of severity, which resulted in a high sensitivity of over 90% and a high specificity of over 90% [86]. Charisis et al. reported an improved algorithm for capsule endoscopy to detect CD lesions with a sensitivity of 95.2%, a specificity of 92.4%, and an accuracy of 93.8% [129]. In a more recent study, Klang et al. employed a deep learning algorithm with more training images for detecting CD lesions by AI-aided capsule endoscopy and demonstrated excellent performance with an AUC of 0.99 and an accuracy of 95.4–96.7% [130]. CT and MRI images are necessary to determine the disease activity in IBD. Although it is challenging for AI to recognize the intestinal wall structure on CT and MRI, semi-automated AI-aided systems have been reported and summarized previously [131,132].

UC-associated dysplasia and cancer are often difficult to detect. A recent case report suggested the usefulness of AI-based colonoscopy for the detection of dysplasia in patients with longstanding UC [133].

## 9. Irritable Bowel Syndrome (IBS)

The prevalence of IBS is estimated at 10–20% worldwide [134]. A few AI-related studies for IBS have been published. Most patients with IBS identify certain foods as triggers for their symptom flare-ups. There are two unique smartphone applications for identifying potential trigger foods. Using photos of food from the mobile applications, Chung et al. developed a personal informatics system, which allows patient–provider collaboration and supports precise individual management [135]. Zia et al. designed an application using an AI algorithm based on regression analyses to identify possible relationships between foods and IBS symptoms. Their two-week study featured assessments of symptoms four times a day and at every meal using a 100-point graded sliding scale [136]. These AI-aided mobile applications tether patients directly to clinicians by capturing frequent and continuous data from patients, and providing individual precision feedback from clinicians to patients. This direct interaction is an advantage of AI and will change health care strategies.

In IBS, gut microbiota is likely linked to its symptoms and pathogenesis [137]. Fukui et al. established a unique AI prediction model for identifying IBS patients based on gut microbiota (sensitivity >80% and specificity >90%) [138].

## 10. Liver Diseases

This section reviews AI-aided image analyses for diagnosing liver masses. In addition, many data analysis studies using AI algorithms have been conducted to predict patients’ outcomes and to discover biomarkers.

### 10.1. Liver Masses

The risk factors for hepatocellular carcinoma (HCC), such as obesity, type 2 diabetes, and nonalcoholic fatty liver disease, are replacing viral- and alcohol-related liver disease [139]. With an increase in metabolic disorders, liver cancer is steadily growing and is the third leading cause of cancer-related death [43,140,141]. 

The detection and diagnosis of liver masses is performed by ultrasonography, CT, and MRI, and AI has been developed for hepatic mass identification. Yasaka et al. employed an AI-aided enhanced CT, which resulted in high performance (AUC = 0.92) in differentiating malignant liver masses (HCCs and other malignant masses) from benign tumors (hemangiomas) or cysts [142]. An AI-aided, multi-phasic MRI developed by Hamm et al. demonstrated higher performance than two radiologists for the detection of six common liver masses (HCC, cyst, hemangioma, focal nodular hyperplasia (FNH) intra-hepatic cholangiocarcinoma, and metastatic tumor) with a sensitivity of 90% vs. 80%/85% and a specificity of 98% vs. 96%/96% [143]. In particular for HCC, the AI had a sensitivity of 90%, compared to 60%/70% from the radiologists. Furthermore, the AI processing speed was extremely fast at 5.6 ms. These results are promising, and the FDA recently approved a liver AI for liver lesion detection by AI-aided MRI and CT (Table 1). It is difficult to develop AI-aided ultrasonography because of several technical issues, which include variability in the data formats and investigator skill level, and, as such, the quality of an ultrasonographic image is highly operator dependent. Although the conditions of examination directly affect the quality of ultrasonographic images, several positive results have been reported and summarized [144]. Schmauch et al. showed that AI-aided ultrasonography detected and diagnosed liver masses (HCC, hemangioma, metastasis, cysts, and FNH) with high performances (AUC 0.935 and 0.916, respectively) [37] (Figure 3D). Enhanced ultrasonography [145] for AI-aided EUS also demonstrated the capability of an EUS-CNN model to autonomously identify liver masses and to accurately classify them as either malignant or benign lesions [146]. AI development in the field of ultrasonography has challenges, including a high dependence on operator experience for acquiring quality imaging data, numerous different equipment vendors and models, multiple image quality parameters, and a high diversity of images and hurdles in database construction. In particular, ultrasound waves require high-speed processing. For histopathology, Sun et al. reported the first paper showing a method to classify liver cancer histopathological images using AI [147]. 

To screen high-risk patients for the development of HCC from patients with cirrhosis, Singal et al. used an AI algorithm and reported good performance [148]. Another important clinical issue for HCC patient management is to identify patients at high risk for post-treatment recurrence. To predict post-operative recurrence, Feng et al. used an AI-aided contrast-enhanced MRI and reported an AUC of 0.83, a sensitivity of 90%, a specificity of 75%, and an accuracy rate of 84% compared to radiologists with an AUC of 0.47–0.57, a sensitivity of 19.3–45.2%, a specificity of 67.3–83.7%, and an accuracy rate of 58.8% [149]. Abajian et al. also showed the utility of AI combined with MRI and patient data [150]. For a similar purpose, Saillard et al. used histopathology images and highlighted the importance of pathologist–AI interactions in the construction of deep-learning algorithms, which benefit from expert knowledge [151]. It was superior to the existing prognostic factors. Factors reflecting a poor prognosis include the presence of vascular space in the tumor and a cord-like shape. AI ultrasonography can also be used for the prediction of response to transcatheter arterial chemoembolization (TACE) and the prediction of post-radiofrequency ablation (RFA) and post-operation survival [152,153].

### 10.2. Nonalcoholic Fatty Liver Disease (NAFLD)

With the increase in systemic metabolic diseases (obesity, diabetes, hyperlipidemia, etc.), the incidence of NAFLD is also increasing worldwide [154]. Since NAFLD-derived HCC is increasing, the early detection of NAFLD is critical to avoid future carcinogenesis. Recently, deep learning algorithms, such as CNNs, have improved the detection of fatty liver disease by ultrasonography [155,156]. Fibrosis is an advanced stage of liver steatosis and the most important risk factor for carcinogenesis. The gold standard for the diagnosis of fibrosis is a liver biopsy, which is invasive and costly [157,158]. The systematic review by Decharatanachart et al. suggested that AI-aided systems (ultrasonography, elastography, CT, and MRI) have promising potential for the diagnosis of liver steatosis and fibrosis with an overall sensitivity of 97% and a specificity of 91% [159]. Elastography is currently the most commonly used modality for staging liver fibrosis [160], and two papers have demonstrated the utility of AI-aided elastography to detect liver fibrosis [161,162]. Gatos et al. designed an AI-aided shear-wave elastography based on a support vector machine model to discriminate chronic liver disease patients (fibrosis) from healthy individuals with a sensitivity of 93.5%, a specificity of 81.2%, an accuracy of 87.3%, and an AUC of 0.87 [162]. Wang et al. applied deep learning to shear wave elastography and compared the AI elastography to a liver biopsy [161]. The AI elastography similarly diagnosed cirrhosis (AUC 0.97) and advanced fibrosis (AUC 0.98). 

Other AI approaches using clinical and laboratory variables routinely measured in clinical practice have been developed [163]. Using serial laboratory data over a person’s timeline, AI analysis can provide a better understanding of a multitude of mechanisms and relationship of risk factors and symptoms. Furthermore, the risk assessment of NAFLD by AI algorithms using serial laboratory variables over a person’s timeline should improve a physician’s management and a patient’s motivation. There are many algorithms challenged in medical AI fields [164], and choosing the best algorithm is an important issue for data analysis by AI. Ma et al. used the Bayesian network model and showed better performance in diagnosing NAFLD based on clinical data than that of logistic regression [165]. Sowa et al. suggested that random forest and decision tree are better than a support vector machine for the separation of NAFLD from alcoholic liver disease [166]. 

### 10.3. Viral Hepatitis

Viral hepatitis (B and C) is still recognized as a major cause of liver cirrhosis and carcinogenesis worldwide, particularly in developing countries. Several AI-based models have been developed to predict the risk of hepatitis-related cirrhosis [167,168,169,170,171]. More recently, a unique prediction model using gut microbiome data was published [172]. Oh et al. used a random forest-based AI algorithm with differential abundance analysis to profile the gut microbiota and metabolites and detect cirrhosis with an AUC of 0.91.

### 10.4. Primary Sclerosing Cholangitis (PSC)

PSC lacks effective medical treatments and occasionally requires a liver transplant due to advanced fibrosis [173]. Moreover, PSC is a premalignant condition and is associated with bile duct cancer at an incidence of 10–30% [173]. Eaton et al. developed an AI-based prediction model, called the Primary Sclerosing Cholangitis Risk Estimate Tool (PREsTo), and demonstrated that the model accurately predicts liver failure in PSC patients, which exceeded the performance of other established, noninvasive prognostic scoring systems [174]. 

### 10.5. Liver Transplantation

Liver transplantation offers an excellent outcome for several end-stage liver disorders. However, challenges remain, such as insufficient donors, high mortality on the waiting list, and graft failures. Regarding the discrepancy between the number of donors and the number of recipients, the appropriate organ allocation should be performed to avoid human bias. The current allocations are based on widely used scoring systems, such as the model for end-stage liver disease (MELD) score, the Delta-MELD score, and the balance-of-risk score, and may yield conflicting results [175,176]. Some AI-based, donor–recipient matching models have been developed [177,178]. Graft failure is the most common problem after liver transplantation. AI-based algorithms developed by Lau et al. using donor, transplant and recipient characteristics predicted graft failure with a high AUC of 0.818 [179]. To identify novel factors associated with death after transplantation, AI has been applied [180,181]. Using a machine learning approach, Bhat et al. found that new-onset or preexisting diabetes was associated with high mortality [180]. 

## 11. Pancreatic Disease

This section reviews AI-aided image and data analyses for the diagnosis of pancreatic disease. 

### 11.1. Pancreatic Cancer

Pancreatic cancer is the seventh most lethal cancer worldwide [43,182]. Tumor size is the most prognostic factor in pancreatic cancer [183]. The five-year survival of patients with lesions smaller than 10 mm (TS1a) is more than 80%, while the five-year survival of patients with larger lesions (>10 mm) is less than 50% [184]. The challenges for pancreatic ductal cancer include a lack of definition in the high-risk group and difficulty in early detection by imaging. Pereira et al. nicely summarized the literature regarding early detection by AI technology [185]. Although abdominal CT is commonly used for screening pancreatic cancer, the detection sensitivity is not high for small lesions [186,187]. To resolve this issue, Liu et al. first trained an AI algorithm with 436 CT images, including 300 normal cases and 136 pancreatic cancer cases [188]. The AI system achieved a sensitivity of 80.2% with a specificity of 90.2%, which may be improved by a larger number of training images. Alternatively, EUS is a more powerful modality to detect small lesions in the pancreas [187,189]. Tonozuka et al. published a pilot study using video to detect pancreatic ductal cancer by AI-based EUS [190]. The system was trained with 920 images of cancers as well as control images from patients with chronic pancreatitis and those with a normal pancreas and, subsequently, validated with an additional 470 test images. The system diagnosed cancers successfully with an AUC of 0.94. To differentiate between cancer and non-cancer (chronic pancreatitis and a normal pancreas), several algorithms have been applied since the first report using a simple conventional algorithm by Norton et al., and three recent reports have used deep learning algorithms [191]. 

The identification of high-risk individuals is another important factor for the early detection of pancreatic cancer [185]. Using AI methodologies and the National Health Insurance Research Database of Taiwan (total 1,358,634 patients), Hsieh et al. developed the first prediction models for pancreatic cancer in patients with type 2 diabetes [192]. They demonstrated that a logistic regression algorithm predicted pancreatic cancer more accurately (AUC of 0.727) than an artificial neural network algorithm, although several researchers have reported that artificial neural networks are suitable to predict some diseases [193]. Further investigations are necessary to identify the most suitable model.

### 11.2. Intraductal Papillary Mucinous Neoplasm (IPMN)

Pancreatic cystic lesions, particularly IPMN, are the precursors of pancreatic cancer [194]. Kuwahara et al. successfully established an AI-aided EUS using deep learning to distinguish malignant IPMNs from benign ones [195]. The AI-aided EUS could diagnose malignant probability with a high sensitivity of 95.7% and a high accuracy of 94.0%, which was much greater than that of experts’ diagnoses (56.0%). AI-aided diagnosis is under development not only for IPMNs but also for other cystic lesions of the pancreas, such as serous cystic neoplasms, mucinous cystic neoplasms, solid pseudopapillary neoplasms, and cystic pancreatic neuroendocrine neoplasms [196].

### 11.3. Autoimmune Pancreatitis (AIP)

Mass-forming AIP may be misdiagnosed as pancreatic cancer and unnecessary surgical resections can occur. Marya et al. demonstrated that an AI-aided EUS accurately differentiated AIP from pancreatic ductal adenocarcinoma and benign pancreatic conditions, thereby permitting an earlier and more accurate diagnosis [197]. The use of this model offers the potential for more timely and appropriate patient care and an improved outcome.

## 12. Future Needs and Conclusions

AI technologies in the medical field hold tremendous promise, although systematic reviews have not provided sufficient evidence that AI outperforms physicians [198]. Several AI-aided devices are commercially available (Table 1), and for future use, multiple studies are on-going in promising areas, such as the identification of anatomical structures and lesions during endoscopic ultrasound, robotic endoscopic surgery, and mobile application. However, there are potential pitfalls, including technical and legal issues [199]. To improve the accuracy of AI diagnosis, more data, including imaging and clinical data, are required to train AI systems. The training data should be collected not only from patients with disease but also from healthy individuals, because larger databases will increase the specificity of the AI system. Particularly for rare diseases, international multicenter projects and open-source libraries, such as ImageNet and cloud net systems [135,136,200], are ideal to provide sufficient training data. However, another issue involves ‘data formatting’ such that different institutions/software may have different data formats. Standardization is critical for future AI developments. To resolve these issues, clinicians need to better understand AI technologies through reading AI-related articles and through collaboration with AI engineers. Even with a large amount of training data, the performance of a particular AI system changes with each training step (annotation, selection of algorithm, selection of data set, etc.), and the addition of inappropriate data will adversely affect performance. Moreover, even in situations where sufficient high-quality training data are used, “overfitting” may occur. To design precise AI systems, we must validate the systems in real-world situations [104,201,202]. 

In conclusion, there is little doubt that AI technology will benefit almost all medical personnel, ranging from specialty physicians to paramedics, in the future [7]. Furthermore, patients should benefit from AI technology directly via mobile applications [135,136]. Physicians should collaborate with the different stakeholders within the AI ecosystem to provide ethical, practical, user-friendly, and cost-effective solutions that reduce the gap between research settings and applications in clinical practice. Collaborations with regulators, patient advocates, AI companies, technology giants, and venture capitalists will help move the field forward.

## Figures and Tables

**Figure 1 diagnostics-11-01719-f001:**
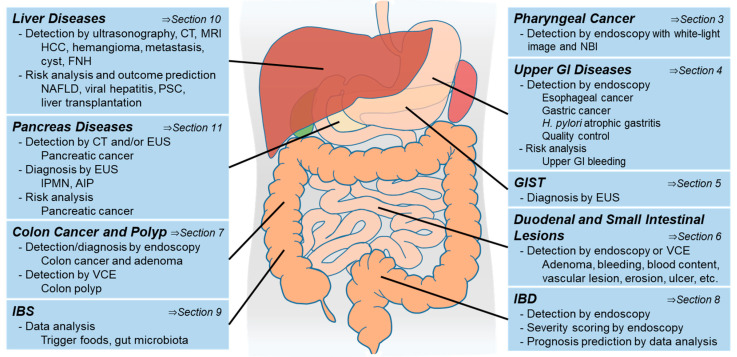
Summary of AI technologies for gastroenterology, hepatology, and pancreatology. IBS: irritable bowel disease, GI: gastrointestinal, GIST: gastrointestinal stromal tumor, IBD: inflammatory bowel disease, EUS: endoscopic ultrasonography, VCE: video capsule endoscopy, NBI: narrow-band imaging, CT: computed tomography, MRI: magnetic resonance imaging, HCC: hepatocellular carcinoma, FNH: focal nodular hyperplasia, IPMN: intraductal papillary mucinous neoplasm, NAFLD: nonalcoholic fatty liver disease, PSC: primary sclerosing cholangitis, AIP: autoimmune pancreatitis.

**Figure 2 diagnostics-11-01719-f002:**
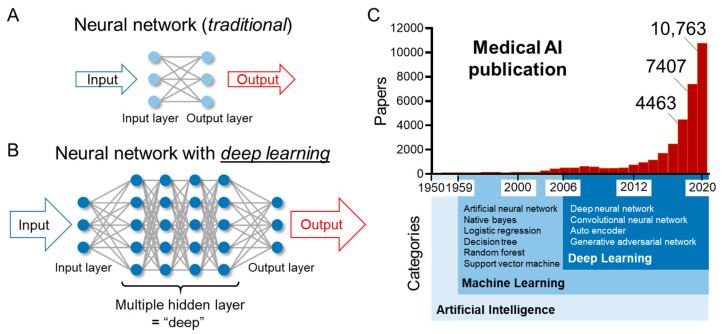
Schematics of neural networks and number of publications of medical AI. (**A**) A schematic of a traditional neural network algorithm. (**B**) A schematic of a neural network with deep learning algorithm. (**C**) The number of publications involving AI in the medical field. The results of a PubMed search using the following key words (“artificial intelligence” OR “machine learning” OR “deep learning” OR “neural network”) AND (medicine OR gastroenterology OR hepatology OR pancreatology OR endoscopy OR radiology OR ultrasonography OR “computed tomography” OR “clinical imaging“) are shown.

**Figure 3 diagnostics-11-01719-f003:**
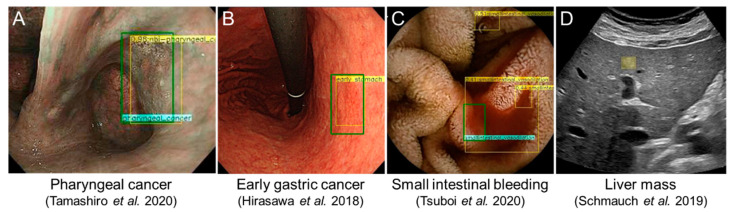
Representative images of AI-aided endoscopies and ultrasonography. (**A**) Pharyngeal cancer detected by AI with narrow-band imaging. Adapted from (Tamashiro A, Dig Endosc 2020) [32]. (**B**) Early gastric cancer detected by an AI system. Adapted from (Hirasawa T, Gastric Cancer 2018) [35]. (**C**) Small intestinal bleeding detected by an AI system. Adapted from (Tsuboi A, Dig Endosc 2020) [36]. (**D**) A liver mass detected by an AI system. Adapted from (Schmauch B, Diagn Interv Imaging 2019) [37].

**Table 1 diagnostics-11-01719-t001:** AI-aided devices approved in the fields of gastroenterology.

Modality	Device Name	Institution	Memo
Endoscopy	EndoBRAIN-EYE	Olympus	Colon tumor detection; made for endocytoscope
	EndoBRAIN	Olympus	Colon tumor diagnosis; made for endocytoscope
	EndoBRAIN-Plus	Olympus	Tumor depth diagnosis; made for endocytoscope
	EndoBRAIN-UC	Olympus	UC activity diagnosis; made for endocytoscope
	CAD EYE	Fujifilm	Colon polyp detection and diagnosis
	WISE VISION	NEC	Colon tumor detection Connectable to 3 major endoscope manufactures
	WavSTAT4	PENTAX ^1^	Colorectal cancer diagnosis
	GI Genius	Medtronic	Colorectal cancer diagnosis
	Discovery	PENTAX ^1^	AI-assisted colon polyp detector
CT	Liver AI	Arterys	Liver lesion detection
US	Poseidon Ultrasound	BUTTERFLY NETWORK	Liver lesion detection

^1^ Hoya group. UC: ulcerative colitis, CT: computed tomography, US: ultrasonography.

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
