# Peer review of "A New Dawn for the Use of Artificial Intelligence in Gastroenterology, Hepatology and Pancreatology"

_diagnostics, 2021, doi:10.3390/diagnostics11091719_

Round 1

Reviewer 1 Report

Congratulations. Excellent review. Well-written, comprehensive and educational. I just have a few observations

Item 9: IBS. This is an incredibly common disorder and any method to provide patient comfort is important. The food analysis by pictures is interesting, but the most important element to this section, IMO, is that AI is tethered directly to the patient, not supplied through a specific image or location (hospital, data bank). Direct patient contact is where the first level AI triage will change health care.

You mention other applications for AI in lines 480-. Lab analysis, especially serial labs over a person’s timeline.

565: AI technologies in the medical field hold tremendous promise, although systematic reviews have not provided sufficient evidence that AI outperforms physicians. ….Evidence based medicine is the goal. AI solutions should be part of the effort to provide more than just image and big data analysis. After all, this is about human care. Disease prevention and treatment in people, not just a sterile computer program owned by a company with a profit-driven stock price.

Author Response

We truly appreciate your comments. Please see our modifications below according to your suggestions. 

Item 9: IBS. This is an incredibly common disorder and any method to provide patient comfort is important. The food analysis by pictures is interesting, but the most important element to this section, IMO, is that AI is tethered directly to the patient, not supplied through a specific image or location (hospital, data bank). Direct patient contact is where the first level AI triage will change health care. > According to your suggestion, we changed the sentence to “These AI-aided mobile application are tethered patients directly to clinicians by capturing frequent and continuous data from patients, and providing individual precision feedback from clinicians to patients. This direct interaction is an advantage of AI and will change health care strategy.”

You mention other applications for AI in lines 480-. Lab analysis, especially serial labs over a person’s timeline. > According to your suggestion, we modified the sentence to “Using serial laboratory data over a person’s timeline, AI analysis can provide better understanding of a multitude of mechanisms and relationship of risk factors and symptoms. Furthermore, the risk assessment of NAFLD by AI algorithms using serial laboratory variables over a person’s timeline should improve physician’s management and patient’s motivation.”

565: AI technologies in the medical field hold tremendous promise, although systematic reviews have not provided sufficient evidence that AI outperforms physicians. ….Evidence based medicine is the goal. AI solutions should be part of the effort to provide more than just image and big data analysis. After all, this is about human care. Disease prevention and treatment in people, not just a sterile computer program owned by a company with a profit-driven stock price. > According to your suggestion, we added the following sentence in Conclusion section “Furthermore, by direct assessment of patient symptoms via mobile applications should improve individual healthy care directly”.

Thanks again for your kind comments.

Reviewer 2 Report

This is a well-organized review of AI in the field of gastroenterology.

1. It would be better to add specific examples of AI use in the form of cases.

2. The principle of AI was described, but it would be better to describe it in more detail. For example, in the case of image recognition, it would be nice to describe how deep learning progresses and AI develops based on existing data.

3. Although AI is being studied a lot and interest is increasing, it has not yet been put into practical use. We expect expert-level diagnostic ability, but there still seem to be limitations. Describe areas that are highly available in the gastroenterology field in near future and areas that are promising and can be used through future research (e.g., identification of anatomical structures and lesions during endoscopic ultrasound).

Author Response

We truly appreciate your comments. Please see our modifications below according to your suggestions. 

It would be better to add specific examples of AI use in the form of cases. > According to your suggestion, we added representative images of AI-aided devices as Figure 3A-D with figure legend. We have obtained permissions (Only Fig3D is pending: waiting for the journal response).

The principle of AI was described, but it would be better to describe it in more detail. For example, in the case of image recognition, it would be nice to describe how deep learning progresses and AI develops based on existing data. > According to your suggestion, We added the following description in Section 2 Artificial Intelligence, “The layer is composed of a filter that extracts features from the original images to deter-mine the characteristics of the original images where higher level features are extracted from lower level ones: for example, first layer extracts patterns at texture level, second layer extracts patterns at frame level and third layer extracts at shape level... last layer indicates list of parts in original input image. Notably, the filter is automatically created after recognition of the features through learning from the input data”.

Although AI is being studied a lot and interest is increasing, it has not yet been put into practical use. We expect expert-level diagnostic ability, but there still seem to be limitations. Describe areas that are highly available in the gastroenterology field in near future and areas that are promising and can be used through future research (e.g., identification of anatomical structures and lesions during endoscopic ultrasound). > According to your suggestion, we added the following sentence in Conclusion section “Several AI-aided devices are commercially available (Table 1) and for future use, multiple researches are on going in promising areas, such as identification of anatomical structures and lesions during endoscopic ultrasound, robotic endoscopic surgery and mobile application.”; “Furthermore, patients should benefit from AI technology directly via mobile applications”.

Thanks again for your kind comments.

Reviewer 3 Report

This article provides an overview of AI applications are available evidences in the field of gastroenterology, hepatology and pancreatic disease. The review is very well written and informative. Several studies are described in details, which makes the overall review good but lengthy and sometimes repetitive to read. Some of the concepts could be better summarized in a shorter text. Future directions and limitations are discussed. Despite there are many promising results in the AI studies, there is still a significant gap between research settings and applications in clinical practice.

Author Response

This article provides an overview of AI applications are available evidences in the field of gastroenterology, hepatology and pancreatic disease. The review is very well written and informative. Several studies are described in details, which makes the overall review good but lengthy and sometimes repetitive to read. Some of the concepts could be better summarized in a shorter text. > We truly appreciate your comments. Please see our modifications below according to your suggestions. According to your suggestions, we have shorten or merged several sentences - in 4.2 Gastric Cancer “Using another CNN algorithm, Wu et al. demonstrated higher performances in AI group than expert endoscopists (accuracy 92.5 vs 89.7%, sensitivity 94% vs 93.9%, specificity 91% vs 87.3%)”; in 8 Inflammatory Bowel Disease “Several AI-aided UC scoring algorithms trained by unbiased UC imaging data that was linked to histological data demonstrated excellent performances in distinguishing endoscopic remission (Mayo 0-1) from moderate-to-severe disease (Mayo 2-3)”.

Future directions and limitations are discussed. Despite there are many promising results in the AI studies, there is still a significant gap between research settings and applications in clinical practice. > According to your suggestions, we added the following sentence in conclusion section, “Physicians should collaborate with the different stakeholders within the AI ecosystem to provide ethical, practical, user-friendly, and cost-effective solutions which reduce a gap between research settings and applications in clinical practice”.

Thanks again for your kind comments.